# A blended wavefield separation method for seismic exploration based on improved GoogLeNet

ZhiQiang Gan[ID]<sup>☯</sup>, XiangE Sun<sup>☯</sup>*

School of Electronic Information and Electrical Engineering, Yangtze University, Jingzhou, HuBei, China

☯ These authors contributed equally to this work.
* 202073043@yangtzeu.edu.cn

**Data Availability Statement:** The datasets used in our paper can be downloaded from the URL: https://doi.org/10.17605/OSF.IO/VW6A3.

**Funding:** The author(s) received no specific funding for this work.

## Abstract

Simultaneous acquisition is a construction method that has been proposed in recent years to meet the requirements of ultra-large-scale and high-precision seismic exploration. Such method is highly efficient and can significantly reduce exploration costs by saving manpower and material resources, being extensively used in offshore exploration and several foreign seismic exploration projects. The data deblending step is a significant part of the operation of simultaneous acquisition, which directly affects the acquired data quality, and is a key factor for the success of oil and gas exploration. The simultaneous use of multiple seismic sources can generate blended noise with a random distribution in non-shot-gather datasets. However, the useful signal exhibits strong coherence, making it possible to separate the non-used wavefield from the blended data. Although the blended noise is randomly distributed in non-shot-gather datasets, it also has characteristics that are different from normal ambient noise, and its kinematic and dynamical characteristics are almost similar to the useful signal. As such, traditional filtering methods are not applicable, especially in the case of strong background noise. In the present study, simultaneous acquisition was introduced, the principle of data deblending using CNN was analyzed, and a data deblending method based on an improved version of GoogLeNet was established. The experimental results show that the trained network model could quickly and effectively separate the mixed wavefield from blended data, and achieve the expected useful signal.

## Introduction

Seismic exploration is presently the most cost-effective and efficient technology for oil and gas resource exploration. Such method relies on the fundamental principle of generating artificial seismic waves to extract information on the underground structural characteristics. With the progress of oil and gas exploration, the distribution of oil and gas reservoirs with shallow burial, easy implementation of structures and good reservoir performance has essentially been mastered. Meanwhile, for oil and gas resources with deep burial and complex structures, high-precision seismic exploration methods need to be applied in order to facilitate a clearer

**Competing interests:** The authors have declared that no competing interests exist.

understanding. Thus, high-precision seismic exploration methods characterized by "broadband, wide orientation, high density, high efficiency, large channel number" have been gradually applied on a large scale. During traditional seismic exploration operations, only the reflected waves from a single source are acquired for a period of time, which can no longer meet the needs of current exploration operations. Simultaneous acquisition is a new method of efficient seismic exploration that has been developed in recent years [1]. The basic principle of the method is to use seismic instruments that operate in microseismic acquisition mode based on satellite timing technology. Multiple sources are then fired simultaneously and efficiently under specific "time-distance rules" to reduce the acquisition time and exploration costs. However, the short time intervals between different sources results in the phenomenon of wavefield mixing, which greatly reduces the SNR of seismic data and the quality of the section plane. Therefore, the effective separation of blended data generated by multiple sources is a significant part of simultaneous acquisition [2–4]. The existing data deblending methods are mostly based on the continuity characteristics of the useful signal and the random distribution characteristics of blended noise in the non-shot-gather domain. In the method, the wavefield generated by a particular source is treated as the signal of interest, while the wavefields generated by other sources are treated as noise. By removing the noise, the data deblending process can be achieved [5, 6].

Deep learning algorithms were first designed to automatically learn hidden features and relationships in massive datasets, being used for regression, classification, and prediction calculations. As examples, Rowley used a deep learning method for face recognition in 1998 [7], which was introduced into medical diagnosis by Kononenko in 2001 [8]. Amongst the background of the recent developments in big data and artificial intelligence, as well as the significant improvement of computing power supported by GPU hardware, deep learning algorithms have developed rapidly since 2010. As a significant branch of artificial intelligence algorithms, deep learning algorithms have been applied in a wide variety of fields. In 2018, Baardman proposed the use of a CNN network to achieve the separation of blended data, for which blended data and corresponding non-blended data were used as the training datasets [9, 10]. In 2019, Slang used non-blended data to generate training datasets, which verified that the deblending effect in the shot-gather domain is sub-optimal, whereas a superior separation solution for mixing wavefields could be achieved in the receiver-gather domain [11]. Richardson realized data deblending by means of a U-Net network based on pre-trained ResNet [12]. In 2020, Nakayama and Sun used a custom U-Net network for the separation of mixing wavefields in blended data in the frequency and time domains respectively, and achieved notable results [13, 14]. Wang proposed a new residual CNN network to deblend data, which continuously iterates to calculate the residuals between the blended data and the predicted data to achieve the purpose of training CNN network for data deblending [15]. Baadman presented a classification network designed to recognize the blended wavefield. Additionally, a deblend CNN network was proposed. Effective data deblending could be achieved after the blended wavefield was properly classified [16]. According to the characteristics of blended noise, Matharu, Zu and Sun proposed that data should be normalized and divided into data blocks with small size when using CNN network for data deblending [17–19], which will reduce the computer hardware requirements. Such methods can solve the problems of complex parameter adjustment, low operation efficiency and high hardware configuration requirements in traditional methods such as direct filtering, sparse inversion and dictionary learning,etc. In the present study, based on the introduction of simultaneous acquisition, the technical principle of using a CNN network to achieve data deblending was analyzed in-depth, and an improved GoogLeNet network based data deblending strategy was constructed, which can provide technical support for further large-scale application of ultra-efficient simultaneous acquisition.

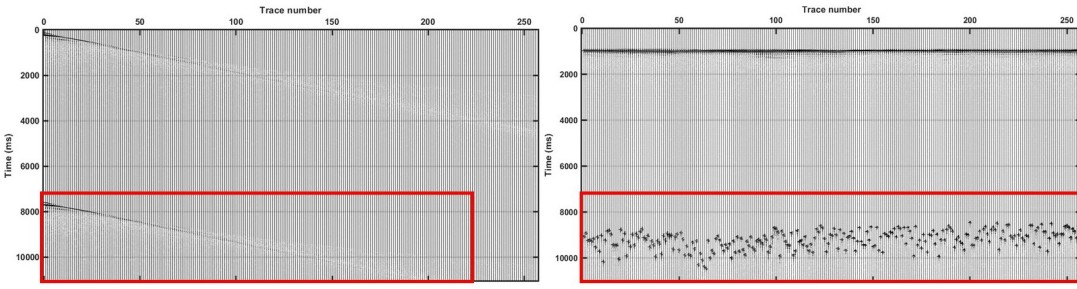

**Fig 1. Different gather displays of blended data.**

## Characteristics and matrix representation of the blended wavefield

### Blended wavefield characteristics

The shot-gather data and trace-gather data display of blended data are shown in Fig 1. The position of the blended noise that needs to be separated is marked with a red line.The left of Fig 1 shows the shot gather. Obviously, the waveform characteristics of the useful data and blended noise data were not much different, which is not conducive to the usage of CNN to extract useful signal features and cannot establish an effective wavefield separation network. The right of Fig 1 shows the trace gather, and the blended waveform shows strong incoherence characteristics, which is obviously different from the waveform characteristics of the useful data. Therefore, a CNN could be selected to deblend the blended wavefield in the trace-gather domain.

### Matrix representation of the blended wavefield

In 1982, Berkout proposed that 2D or 3D seismic data could be described as denoted in Eq (1). Here, $z_d, z_s$ are related to the depth of receiver and source point, respectively; $P(z_d, z_s)$ denotes the recorded seismic data matrix associated with a single source; $D(z_d)$ denotes the receiver points matrix; $X(z_d, z_s)$ denotes the multidimensional transfer function of the subsurface structure; and $S(z_s)$ denotes the source points matrix.

$$P(z_d, z_s) = D(z_d)X(z_d, z_s)S(z_s) \tag{1}$$

In general, the simultaneous acquisition can be formulated as shown in Eq (2).

$$P'(z_d, z_s) = D(z_d)X(z_d, z_s)S'(z_s) \tag{2}$$

where $P'(z_d, z_s)$ and $S'(z_s)$ are the blended source matrix and the blended data matrix.

$$S'(z_s) = S(z_s)\Gamma(z_s) \tag{3}$$

In Eq (3), $\Gamma(z_s)$ is the blending matrix, which contains the blending parameters. By substituting Eqs (1) and (3) into Eq (2), the following could be obtained:

$$P'(z_d, z_s) = P(z_d, z_s)\Gamma(z_s) \tag{4}$$

The matrix expression of Eq (4) is:

$$\begin{bmatrix} p'_{1,1} & \cdots & p'_{m-1,1} & p'_{m,1} \\ p'_{1,2} & \cdots & p'_{m-1,2} & p'_{m,2} \\ \vdots & \cdots & \vdots & \vdots \\ p'_{1,n-1} & \cdots & p'_{m-1,n-1} & p'_{m,n-1} \\ p'_{1,n} & \cdots & p'_{m-1,n} & p'_{m,n} \end{bmatrix} = \begin{bmatrix} p_{1,1} & \cdots & p_{k-1,1} & p_{k,1} \\ p_{1,2} & \cdots & p_{k-1,2} & p_{k,2} \\ \vdots & \cdots & \vdots & \vdots \\ p_{1,n-1} & \cdots & p_{k-1,n-1} & p_{k,n-1} \\ p_{1,n} & \cdots & p_{k-1,n} & p_{k,n} \end{bmatrix} * \begin{bmatrix} \gamma_{1,1} & \cdots & \gamma_{m-1,1} & \gamma_{m,1} \\ \gamma_{1,2} & \cdots & \gamma_{m-1,2} & \gamma_{m,2} \\ \vdots & \cdots & \vdots & \vdots \\ \gamma_{1,k-1} & \cdots & \gamma_{m-1,k-1} & \gamma_{m,k-1} \\ \gamma_{1,k} & \cdots & \gamma_{m-1,k} & \gamma_{m,k} \end{bmatrix} \quad (5)$$

where $p'_{m,n}$ denotes the response of $m$ times blended shooting at $n$ receiver point; $p_{k,n}$ denotes the response of $k$ times unblended shooting at $n$ receiver point; and $\gamma_{m,k}$ denotes the encoding factor of the $k$ unblended source in $m$ times shooting.

Due to the nonlinearity of the blending encoding method, when only certain sources are blended and not all sources, certain elements in the encoding factor matrix are rendered as zero. Consequently, the corresponding equations in Eq (5) become underdetermined, and a unique solution cannot be obtained. Specifically, $\Gamma^{-1(z_s)}$ does not exist, which is also a problem to be solved in the data deblending process. The optimal solution is usually obtained by iterative calculation. However, such method has high requirements in terms of computer hardware configuration and complex parameter setting.

## Blended wavefield separation method

### Algorithmic model

Convolutional neural networks (CNNs) generally include basic units such as a convolution layer, normalization layer and activation layer. As a kind of feedforward neural network with a certain depth, CNNs have strong capabilities in respect of image feature recognition and extraction. At present, CNNs are widely used in the field of digital image processing and have significant advantages. The seismic data processing has many similarities with digital image processing. Thus, a CNN for digital image processing could be selected to solve the underdetermined problem of Eq (5), and realize the wavefield separation of the blended data. The matrix on the right side of Eq (4) could be further decomposed into the useful data matrix $P_{data}$ and the blended noise matrix $P_{noise}$:

$$P'(z_d, z_s) = P_{data}(z_d, z_s) + P_{noise}(z_d, z_s) \quad (6)$$

The relationship between $P_{data}(z_d, z_s)$ and $P'(z_d, z_s)$ based on a CNN is shown in Eq (7), where $\Theta$ represents the network parameters. After training based on a large amount of label data, the CNN could be used to directly obtain the data deblending result $P_{data}$, or blended noise data $P_{noise}$. Subsequently, the $P_{data}$ could be obtained after simple calculation, as shown in Fig 2. For simplicity, only the method of directly obtaining $P_{data}$ was explored in the present study, which is Method 1 in Fig 2.

$$P_{data} = NET(P', \Theta) \quad (7)$$

GoogLeNet is a new deep learning network model that was proposed by Christian Szegedy in 2014 [20]. Said model has the capacity to significantly enhance network training effectiveness by extracting more features while utilizing the same computational resources, achieved through the implementation of the modular inception structure. In our study, a new deep network was obtained by improving the GoogLeNet network structure, as shown in Fig 3. To

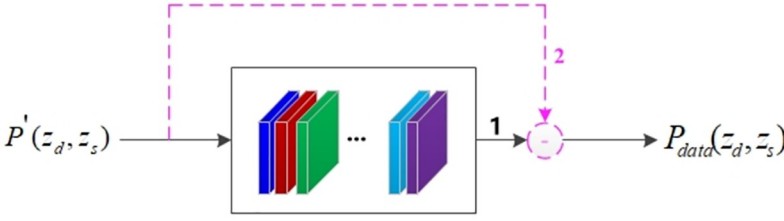

**Fig 2. Schematic diagram of data deblending method.**

achieve better results in deblending process, this study optimized the network structure based on GoogLeNet. Firstly, the number of input channels was reduced from 3 to 1, and the network output type was changed from classification to regression. Secondly, the receptive field of the model's first convolutional layer was adjusted from 7×7 to 3×3, and the 5×5 convolution operation channel in Inception model was removed, while adding 3 transpose convolutional layers after 5 Inception layers for data upsampling. Additionally, while maintaining the network's ability to recognize blended wavefields and unblended wavefields, a pooling operation unit used for input data downsampling in GoogLeNet was removed, significantly reducing the hardware requirements for algorithm implementation.

## Data conversion method

As aforementioned, deep learning was first applied in the field of image processing, and the corresponding network model is more suitable for digital data. However, the data characteristics of seismic data are considerably different from those of pictures. The primary indication is that the actual value of samples exhibits fluctuations ranging from microvolts to volts, indicating a significant dynamic range that may impact the convergence of network training and the effectiveness of data deblending. In order to apply the deep learning algorithm to the separation of blended noise, the range of seismic data needs to be converted to render data features

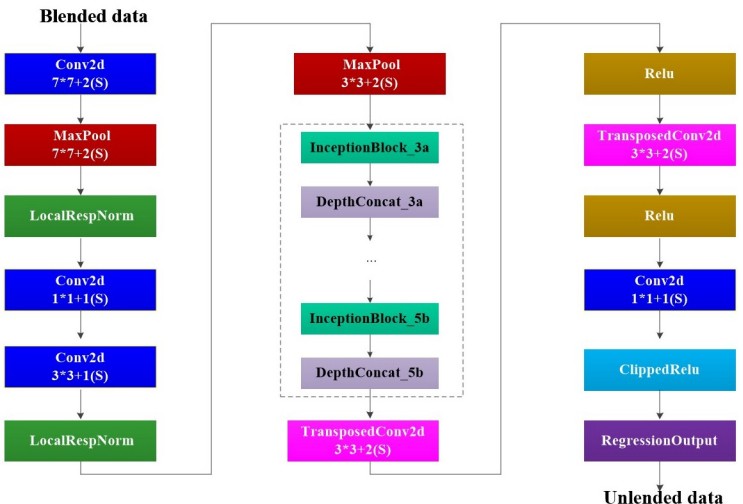

**Fig 3. Schematic diagram of data deblending method.**

that can better match the deep learning network algorithm. The conversion calculation method is shown in Eq (8).

$$X'(t) = \frac{X(t) - min}{max - min} * value, value \in [1, 255] \tag{8}$$

where $X(t)$ denotes the seismic data value before conversion; $min$ and $max$ are the minimum and maximum values of seismic data, respectively; and $X'(t)$ represents the converted data value, with the numerical range being (0, $value$).

After using the deep learning network to complete the training process and produce the deblending results, the inverse process of the calculation process shown in Eq (8) needs to be used to restore the predicted data and obtain the final data deblending results, as shown in Eq (9).

$$X_p(t) = \frac{(max - min) * P(t)}{value} + min, value \in [1, 255] \tag{9}$$

where $P(t)$ denotes the predictive output of the deep network; and $X_p(t)$ is the data deblending result. The $min, max$ and $value$ are the same as those in Eq (8).

## Quantitative evaluation

The purpose of data deblending is to obtain the useful data from the blended data and remove the blended noise, which is similar to digital image denoising. In order to evaluate the data deblending effect more intuitively, the root mean squares error (RMSE), signal to noise ratio (SNR) and peak signal to noise ratio (PSNR) were used as quantitative evaluation indexes, and could be defined as follows:

$$RMSE = sqrt\left(\frac{1}{N*L}\sum_{i=1}^{N}\sum_{j=1}^{L}[P_{data}(i,j) - P_{theory}(i,j)]^2\right) \tag{10}$$

where $N$ is the number of acquisition channels; $L$ is the number of samples acquired by a single channel; $P_{data}(i,j)$ and $P_{theory}(i,j)$ are the deblending results by CNN and the valid signal lablels, respectively. $RMSE$ is used to characterize the separation degree of blended noise, wherein the smaller the value, the higher the suppression degree of blended noise.

$$SNR = 10 * lg\frac{\sum_{i=1}^{N}\sum_{j=1}^{L}[P_{theory}(i,j)]^2}{\frac{1}{N*L}\sum_{i=1}^{N}\sum_{j=1}^{L}[P_{data}(i,j) - P_{theory}(i,j)]^2} \tag{11}$$

$$PSNR = 10 * lg\frac{MAX^2}{\frac{1}{N*L}\sum_{i=1}^{N}\sum_{j=1}^{L}[P'(i,j) - P(i,j)]^2} \tag{12}$$

where $MAX$ is the maximum value of the blended data, which is related to the type of sensors used and the preamp gain settings, typically not exceeding 2.5V. The rest of the symbol definitions are the same as those in Eq (10). An observation can be made from Eqs (11) and (12) that the larger the $SNR$ or $PSNR$, the better the protection of the useful signal by the data blending algorithm.

## Data validation

### Experiment environment

The experiment was conducted using the deep learning toolbox of Matlab on a Windows 10 64-bit operating system. The hardware specifications were as follows: an Intel® Xeon® CPU E5-2680 v3 @2.50GHz for the CPU, 32GB of random access memory, and an NVIDIA GeForce RTX 3060 graphics processing unit with 12GB memory size.

The proposed network architecture was designed using the builtin Deep Network Designer toolbox in Matlab 2021a, and the algorithm proposed was implemented using the Matlab language.

### Dataset processing

The dataset includes a blended data matrix and an unblended data matrix. The size of the each data matrix is 2768×256×1×256. The first dimension (of size 2768) corresponds to time samples. The second and fourth dimensions (of size 256) correspond to receiver and source indexes, and the third dimension (of size 1) is reserved. In the dataset, the time intervals between 2 or more sources in a shot-gather record are sometimes reduced to an extent that the time between two sources is less than the time it takes for all reflections caused by a source to dampen. This causes simultaneous sources, where one source is the source of interest and the other source(s) are considered interference.

During the experiment, the dataset processing involved data domain transformation, data range conversion before deblending calculation, data re-conversion after deblending calculation, and dataset partitioning. Firstly, similar to the preprocessing process of seismic data for wavefield deblending calculation, the seismic data to be processed is converted from common shot gather to common receiver gather to enhance the model's ability to perceive the features of the blended and unblended wavefield. Next, according to Eq (8), the data range conversion calculation was applied to the seismic data in the common receiver domain to avoid adverse effects from large data values on model parameter updates and to accelerate training convergence. In the conversion calculation process, the "value" in Eq (8) is 1. Then, the seismic data in the common receiver domain, after undergoing data range conversion, was sliced into 48×96-sized data blocks, and the data was randomly divided into training, validation, and test sets according to the ratio of 8:1:1. After completing the model training, the prediction results are obtained when the seismic data containing the blended wavefield is used as the model input, and then the final deblending results are calculated by Eq (9).

### Test results

In order to verify the blended wavefield separation effect of the proposed model based on improved GoogLeNet, the network models commonly used in deep learning such as AlexNet, VGG-16, VGG-19, UNET and original GoogLeNet are selected for comparative experiments. The parameters comparison of different networks is shown in Table 1. During the training process, the adam optimizer was employed with a batch size of 64, a maximum iteration limit of 60, an training rate of 0.001. The seismic data after preprocesing caculation is input into the above network models for training, verification and testing.

The RMSE curves during the validation process of different network models are shown in Fig 4. After 1000 training iterations, the RMSE curve of the proposed deblending model tends to stabilize, with RMSE values generally lower than GoogLeNet, VGG-16, VGG-19, AlexNet, U-Net. Therefore,the performance of the proposed model in the validation dataset is relatively better.

**Table 1. The paprameters comparision of the different deblending model.**

| Parameters | AlexNet | VGG-16 | VGG-19 | U-Net | GoogLeNet | Proposed model |
|---|---|---|---|---|---|---|
| Layer number | 25 | 33 | 37 | 42 | 144 | 108 |
| Connection number | 24 | 32 | 36 | 44 | 170 | 125 |
| Kernal size of first convolutional layer | 3×3 | 3×3 | 3×3 | 3×3 | 3×3 | 3×3 |
| convolutional layer number | 6 | 11 | 13 | 10 | 58 | 40 |
| Transposed convolutional layer number | 3 | 3 | 3 | 3 | 3 | 3 |

After 60 epochs, we apply the trained network to the test dataset. The original data and its corresponding theoretical deblended solution are shown in Fig 5.

The deblending solutions of different deep learning network are shown in Fig 6. It is evident that the blended noise have been significantly suppressed, and the seismic signal have been well preserved.

To further analyze the effectiveness of different models in separating blended wavefields, a random trace from the common receiver gather was selected for comparison. After deblending caculation, the results of the unblended part and the blended part in the data are shown in Figs 7 and 8 respectively. It can be seen that different models have basically the same effects in suppressing the blended noise, while the proposed model exhibits superior capability in preserving the unblended wavefield compared to the other models.

The quantitative evaluation indexes of different deblending model solutions are shown in Table 2. It can be seen from the table that after 60 training Epochs, the proposed model achieves superior RMSE, SNR, and PSNR values on the test dataset compared to the other neural network models.

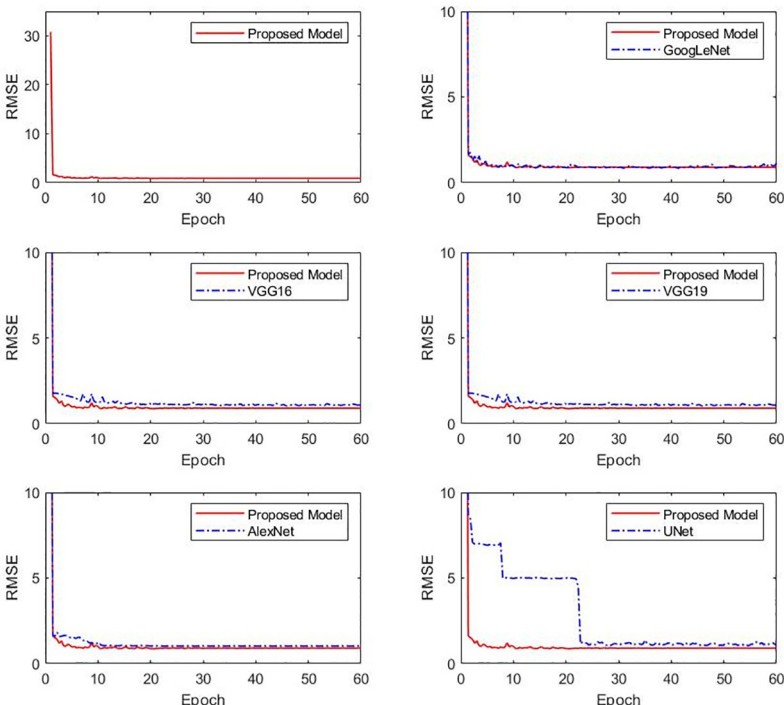

**Fig 4. The validation RMSE comparision of different deblending network during training.**

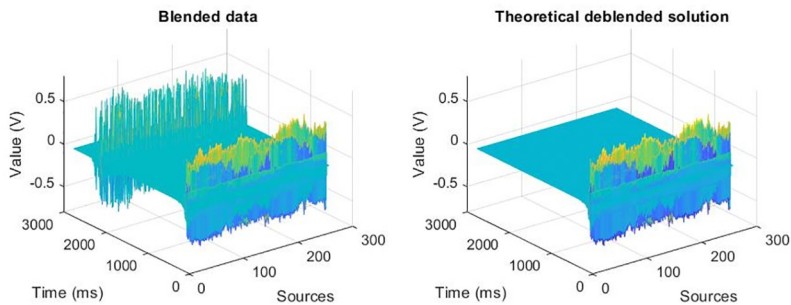

**Fig 5. Display of the original data and its corresponding theoretical deblended solution.**

## Conclusion

The separation of blended wavefields constitutes a crucial component of blended acquisition operations. Typically, the continuity of the useful signal in the non-shot-gather domain of the blended data is exploited to isolate the wavefield generated by the neighboring source as random noise. However, the kinematic and kinetic characteristics of the blended noise closely resemble those of the useful signal. Consequently, traditional filtering algorithms employed in seismic data processing fail to yield satisfactory outcomes, particularly in scenarios where ambient noise is prevalent. While sparse inversion and dictionary learning algorithms have demonstrated good results in specific cases, they necessitate complex parameter configurations, exhibit lower efficiency, longer computation times, and higher hardware requirements compared with filtering theory-based methods [21, 22], which may not be adequate for meeting the demands of high-precision seismic data processing.

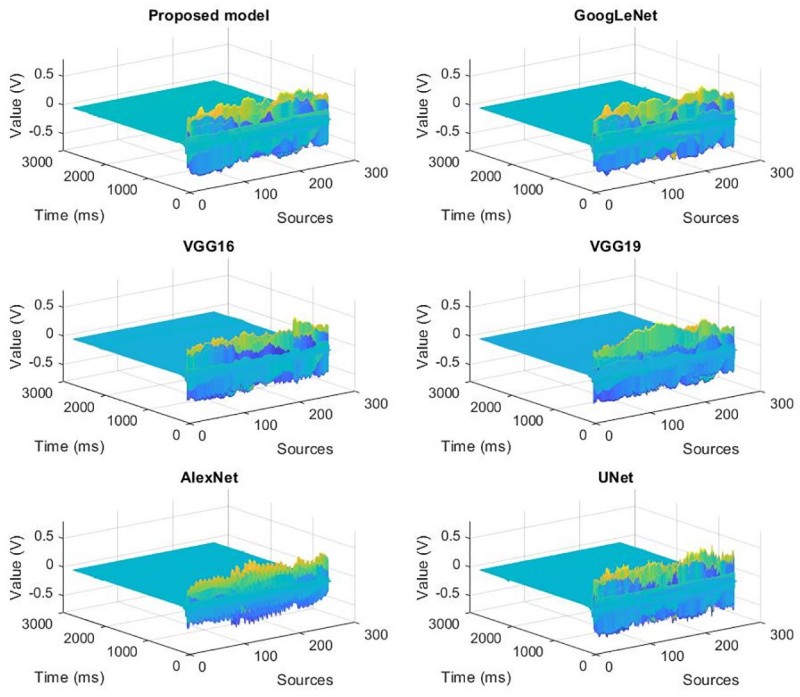

**Fig 6. The solutions of different deep learning network.**

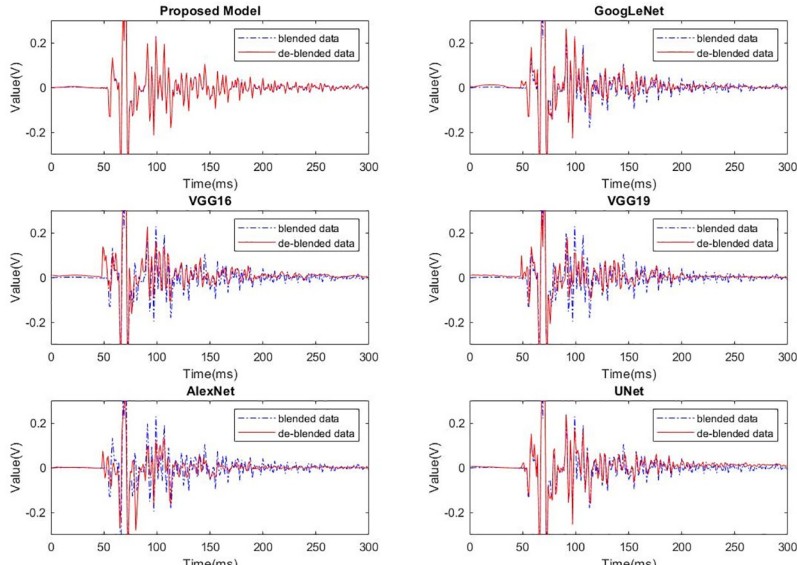

**Fig 7. Data deblending solutions of the unblended wavefield of different models.**

GoogLeNet achieves more efficient utilization of computational resources based on the use of inception module, and is able to extract more detailed features with the same amount of computation. In the present study, based on the theoretical analysis of the feasibility of using deep convolutional neural networks for blended wavefield separation, a deblending model was proposed based on an improved GoogLeNet network. Through data validation experiments, the following conclusions were obtained.

1. Since the seismic data has a large dynamic range, such data needs to be pre-processed before separating the blended wavefield, so that the data characteristics can be better

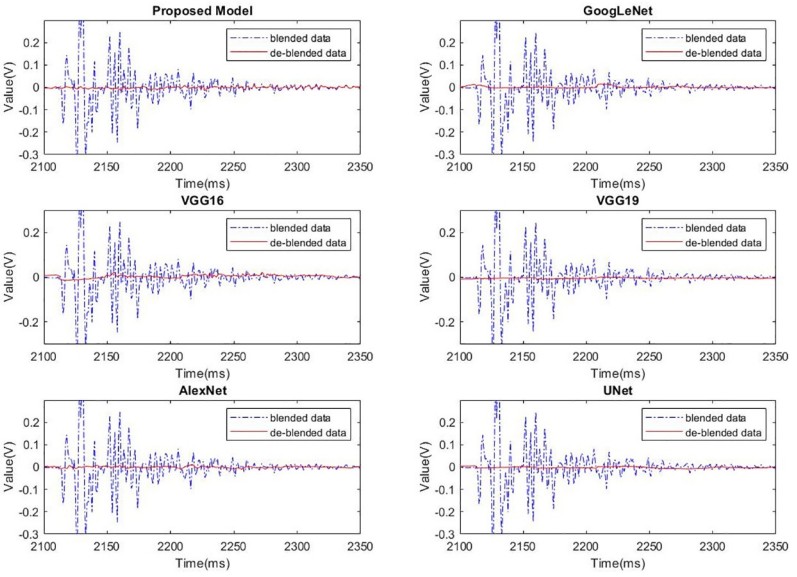

**Fig 8. Data deblending solutions of the blended wavefield of different models.**

**Table 2. The paprameters comparision of the different deblending model.**

| Parameters | AlexNet | VGG-16 | VGG-19 | U-Net | GoogLeNet | Proposed model |
|---|---|---|---|---|---|---|
| RMSE | 6.4123 | 2.8092 | 2.3826 | 1.1631 | 1.0404 | 0.7684 |
| SNR | 4.3728 | 8.7639 | 9.0136 | 13.3821 | 15.0012 | 17.8211 |
| PSNR | 2.8840 | 8.9718 | 7.5411 | 1.3126 | 0.3448 | 16.1403 |

matched with the deep learning algorithm, and then the required processing results can be obtained through the inverse pre-processing process.

2. Due to the large size of the seismic data and its influence on the later data separation results, the hardware configuration of the computer will have higher requirements in network training and the later prediction process, which cannot meet the needs of high precision seismic exploration. Dividing seismic data into data blocks of appropriate size can solve such problem effectively.

3. The deep learning network model can effectively extract detailed features relevant to the problem from the input data blocks. Upon completion of training, the model can effectively separate the blended wavefield. The deblending model proposed in this study, based on the improved GoogLeNet, inherits GoogLeNet's excellent characteristics while simplifying model complexity. It rapidly and accurately extracts relevant detailed features, facilitating effective separation of blended wavefields.

## Author Contributions

**Data curation:** ZhiQiang Gan.

**Methodology:** ZhiQiang Gan.

**Project administration:** XiangE Sun.

**Resources:** XiangE Sun.

**Software:** ZhiQiang Gan.

**Supervision:** XiangE Sun.

**Validation:** XiangE Sun.

**Writing – original draft:** ZhiQiang Gan.

**Writing – review & editing:** ZhiQiang Gan.

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
