## [Decision Letter · Decision Letter 0]

29 Feb 2024

PONE-D-24-02777A blended wavefield separation method for seismic exploration based on deep learningPLOS ONE

Dear Dr. Gan,

Thank you for submitting your manuscript to PLOS ONE. After careful consideration, we feel that it has merit but does not fully meet PLOS ONE’s publication criteria as it currently stands. Therefore, we invite you to submit a revised version of the manuscript that addresses the points raised during the review process.

The reviewers have raised major concerns, Please address all the comments in the revised version.

We look forward to receiving your revised manuscript.

Kind regards,

Alberto Marchisio

Academic Editor

PLOS ONE

3. We note that your Data Availability Statement is currently as follows: [The blended wavefield dataset used in this paper were downloaded from SEG WiKi and are available for public use.]

Reviewers' comments:

Reviewer's Responses to Questions

**Comments to the Author**

1. Is the manuscript technically sound, and do the data support the conclusions?

Reviewer #1: Partly

Reviewer #2: Partly

Reviewer #3: No

2. Has the statistical analysis been performed appropriately and rigorously? 

Reviewer #1: Yes

Reviewer #2: Yes

Reviewer #3: No

3. Have the authors made all data underlying the findings in their manuscript fully available?

Reviewer #1: No

Reviewer #2: No

Reviewer #3: No

4. Is the manuscript presented in an intelligible fashion and written in standard English?

Reviewer #1: Yes

Reviewer #2: Yes

Reviewer #3: No

5. Review Comments to the Author

Reviewer #1: In this paper, the authors proposed “A blended wavefield separation method for seismic exploration based on deep learning”. However, a lot of research has been studied on the application of deep learning methods in the blended seismic data. The title does not reflect the innovative points, but only repeats the results of the previous work. Therefore, this paper does not meet the requirement of the published paper.

In addition, it is suggested that the authors should study the characteristics of blended seismic data, and design targeted neural networks to solve the problem of the blended seismic data.

Reviewer #2: Within the scope of the study, a blended signal decomposition application using deep learning methods for seismic exploration was realized. I find the study generally successful. However, I think that the following points should be taken into consideration within the scope of the study.

1. The source and acquisition procedures of the data used in the study should be included in detail. Was this data taken from an open data set? If it was collected by the study team, where and when was it obtained? Has it been subjected to any pre-exclusion? From how many different sources was it obtained? The article is expected to answer these and similar questions.

2. Considering the limited number of data used in the study, have measures been taken against the problem of overfitting? Is there a possibility that the deep learning network may have memorized?

3. Within the scope of the study, the data set is divided into three as training, validation and testing. Since the dataset used in the study is small, cross-validation is recommended.

4. If data obtained from a single source is used, it is recommended to verify with a different source in order to generalize the results. For example, if two seismic sources are used in the dataset, what will be the result when a blended signal with four sources is encountered?

Reviewer #3: The manuscript is targeting an important topic in seismic exploration survey. Nevertheless, I think the manuscript is not well organized and the novelty, and contents are not sufficient for the journal. I recommend the editor to reject the paper. I summarize my general comments below in case the authors plan to rework the manuscript and submit it to this or other journals:

1- Deblending is a very specific task in seismic exploration workflow. So, maybe it would be better to dig into the task itself rather than a very general perspective about seismic data importance. A big portion of the manuscript is dedicated to the basics of the seismic surveys.

2- Again the authors in paragraph 2 of the introduction talk about the basics of the ML. The manuscript should be sharp and to the point. Any literature review should be relevant to the context and its detail should be useful.

3- The authors start with chronological literature review of the ML for deblending. Why chronological? Do you think each author followed the path of the previous ones. It doesn’t seem that way. The way the papers are cited is completely wrong. Also, at some point (line 46 onward) the authors decided not to continue chronologically, why?

4- The authors keep using misleading phrases. It is constantly mentioned in the manuscript that the GoogLeNet was improved. Do you mean you calibrated/tailored/fine-tuned the GoogleNet for your purpose? A process that is usually referred to as Transfer Learning in ML world is not improvement.

5- This is the first manuscript I have seen that includes two INTRODUCTION chapters.

6- The basic principles section of the second introduction includes very basic principles of blended acquisition. Do you think this is interesting for an audience. There are reference books to learn about blended acquisition.

7- Where is the data chapter? Where does the data come from? Very bad quality pictures with no descriptions. For example, Figure 5, where are the axis. What are the plots?

8- From what I see from the left panel in Figure 5, the data are not even blended. It seems the shootings were not even close in time to cause blending of the wavefields.

9- What is the value in equation 11? Do you mean you normalize the data to gray scale? Your equation does not reflect that.

10- What is the benchmark method? How the deblended data is labeled?

11- The authors say the misfit doesn’t change after 40 epochs. It seems they reach a local minimum much earlier.

12- What is figure 8, 9. No description. No interpretation.

13- In conclusions the authors say: “Consequently, traditional filtering algorithms employed in seismic data processing fail to yield satisfactory outcomes, particularly in scenarios where ambient noise is prevalent.” Where is this information in the paper? I don’t see any comparison. Benchmark method or traditional method.

14- Again, in conclusion, the authors compare the performance of their application with other algorithms that were not shown.

15- In addition to these comments that are summarized the language of the paper is significantly poor. Some of the phrases are misleading and some can not be understood.

6. PLOS authors have the option to publish the peer review history of their article (what does this mean?). If published, this will include your full peer review and any attached files.

Reviewer #1: No

Reviewer #2: No

Reviewer #3: No

---

## [Author Response · Author response to Decision Letter 0]

21 Mar 2024

Reviewer #1: 

1. In this paper, the authors proposed “A blended wavefield separation method for seismic exploration based on deep learning”. However, a lot of research has been studied on the application of deep learning methods in the blended seismic data. The title does not reflect the innovative points, but only repeats the results of the previous work. Therefore, this paper does not meet the requirement of the published paper.

Response:We have modified the title of the article to “A blended wavefield separation method for seismic exploration based on improved GoogLeNet” to provide more appropriateness with the content of the article.

2. In addition, it is suggested that the authors should study the characteristics of blended seismic data, and design targeted neural networks to solve the problem of the blended seismic data.

Response:We thoroughly reorganized and extensively analyzed the characteristics and mathematical representation of blended seismic data, and described the proposed deblending model based on improved GoogLeNet in detail in “Blended wavefield separation method” section of the paper to addressing the concerns raised by Reviewer #1.

Reviewer #2: 

1. The source and acquisition procedures of the data used in the study should be included in detail. Was this data taken from an open data set? If it was collected by the study team, where and when was it obtained? Has it been subjected to any pre-exclusion? From how many different sources was it obtained? The article is expected to answer these and similar questions.

Response: The dataset used in this paper were downloaded from SEG WiKi and are available for public use, which provided the data by Petroleum Geo-Systems (PGS) for the 2017 IEEE signal processing competition in Tokyo (2017). After downloading the dataset, we reorganized the dataset.The reorganized dataset includes a blended data matrix and an unblended data matrix.The size of the each data matrix is 2768×256×1×256. The first dimension (of size 2768) corresponds to time samples. The second and fourth dimensions (of size 256) correspond to receiver and source indexes, and the third dimension (of size 1) is reserved. In the dataset, the time intervals between 2 or more sources in a shot-gather record are sometimes reduced to an extent that the time between two sources is less than the time it takes for all reflections caused by a source to dampen.

The above information has been elaborated in the “data vaditation –dataset processing” part of the article. 

2. Considering the limited number of data used in the study, have measures been taken against the problem of overfitting? Is there a possibility that the deep learning network may have memorized?

Response： Aiming at the limited number of data used in the study, the seismic data is divided into 48×96 sized data blocks before training. On the one hand, the hardware requirements in the training process can be reduced, and on the other hand, the number of training data samples can be increased. At the same time, combined with the characteristics of training samples, the Inception model in GoogLeNet is improved to simplify the model structure and avoid overfitting.

3. Within the scope of the study, the data set is divided into three as training, validation and testing. Since the dataset used in the study is small, cross-validation is recommended.

Response: As mentioned before, the the seismic data is divided into 48×96 sized data blocks before training, and the number of training samples is 71688, the number of validation or testing is 8700.According to Reviewer #2’s suggestion, we compared the training accuracy,validation accuracy and the accuracy of the test data .These values are basically the same. Therefore, the method of cross validation is not used in this paper.

4. If data obtained from a single source is used, it is recommended to verify with a different source in order to generalize the results. For example, if two seismic sources are used in the dataset, what will be the result when a blended signal with four sources is encountered?

Response:The dataset used in this paper were downloaded from SEG WiKi and are available for public use, which provided the data by Petroleum Geo-Systems (PGS) for the 2017 IEEE signal processing competition in Tokyo (2017). After downloading the dataset, we reorganized the dataset.The reorganized dataset includes a blended data matrix and an unblended data matrix.The size of the each data matrix is 2768×256×1×256. The first dimension (of size 2768) corresponds to time samples. The second and fourth dimensions (of size 256) correspond to receiver and source indexes, and the third dimension (of size 1) is reserved. In the dataset, the time intervals between 2 or more sources in a shot-gather record are sometimes reduced to an extent that the time between two sources is less than the time it takes for all reflections caused by a source to dampen.

The above information has been elaborated in the “data vaditation –dataset processing” part of the article. 

Reviewer #3:

1.Deblending is a very specific task in seismic exploration workflow. So, maybe it would be better to dig into the task itself rather than a very general perspective about seismic data importance. A big portion of the manuscript is dedicated to the basics of the seismic surveys.

Response：The introduction section of the paper has been modified to elucidate the industrial demand for blended seismic wavefield separation and the current research status of data deblending by deep learning method, aiming to demonstrate the necessity and significance of our study.

2- Again the authors in paragraph 2 of the introduction talk about the basics of the ML. The manuscript should be sharp and to the point. Any literature review should be relevant to the context and its detail should be useful.

Response ：The introduction section of the paper has been modified to elucidate the industrial demand for blended seismic wavefield separation and the current research status of data deblending by deep learning method, aiming to demonstrate the necessity and significance of our study.

3- The authors start with chronological literature review of the ML for deblending. Why chronological? Do you think each author followed the path of the previous ones. It doesn’t seem that way. The way the papers are cited is completely wrong. Also, at some point (line 46 onward) the authors decided not to continue chronologically, why?

Response :The introduction section of the paper has been modified to elucidate the industrial demand for blended seismic wavefield separation and the current research status of data deblending by deep learning method, aiming to demonstrate the necessity and significance of our study.

4- The authors keep using misleading phrases. It is constantly mentioned in the manuscript that the GoogLeNet was improved. Do you mean you calibrated/tailored/fine-tuned the GoogleNet for your purpose? A process that is usually referred to as Transfer Learning in ML world is not improvement.

Response:We thoroughly reorganized and extensively analyzed the characteristics and mathematical representation of blended seismic data, and described the proposed deblending model based on improved GoogLeNet in detail in “Blended wavefield separation method” section of the paper.

5- This is the first manuscript I have seen that includes two INTRODUCTION chapters.

Response: delete some contents of the second section which named “Basic principle” to simply the paper.

6- The basic principles section of the second introduction includes very basic principles of blended acquisition. Do you think this is interesting for an audience. There are reference books to learn about blended acquisition.

Response: delete some contents of the second section which named “Basic principle” to simply the paper.

7- Where is the data chapter? Where does the data come from? Very bad quality pictures with no descriptions. For example, Figure 5, where are the axis. What are the plots?

Response: The dataset used in this paper were downloaded from SEG WiKi and are available for public use, which provided the data by Petroleum Geo-Systems (PGS) for the 2017 IEEE signal processing competition in Tokyo (2017). We reorganized and uploaded the dataset.The reorganized dataset includes a blended data matrix and an unblended data matrix.The size of the each data matrix is 2768×256×1×256. The first dimension (of size 2768) corresponds to time samples. The second and fourth dimensions (of size 256) correspond to receiver and source indexes, and the third dimension (of size 1) is reserved. In the dataset, the time intervals between 2 or more sources in a shot-gather record are sometimes reduced to an extent that the time between two sources is less than the time it takes for all reflections caused by a source to dampen.

The above information has been elaborated in the “data vaditation –dataset processing” part of the article. 

8- From what I see from the left panel in Figure 5, the data are not even blended. It seems the shootings were not even close in time to cause blending of the wavefields.

Response: In order to verify the blended wavefield separation effect of the proposed model based on improved GoogLeNet, the network models commonly used in deep learning such as AlexNet, VGG-16, VGG-19, UNET and original GoogLeNet are selected for comparative experiments.The parameters of different models, the vadidation RMSEcurves in the training process and the quantitative evaluation index of the de-blending results in the test dataset are also supplemented . Through the detailed comparative analysis of different models in the training process and the performance in the test dataset, the performance and effectiveness of the proposed blended wavefield separation model based on improved GoogLeNet are further validated.

9- What is the value in equation 11? Do you mean you normalize the data to gray scale? Your equation does not reflect that.

Response: In the data conversion calculation process, the "value" in the equation is 1.

10- What is the benchmark method? How the deblended data is labeled?

Response: The dataset used in this paper were downloaded from SEG WiKi and are available for public use, which provided the data by Petroleum Geo-Systems (PGS) for the 2017 IEEE signal processing competition in Tokyo (2017). We reorganized and uploaded the dataset.The reorganized dataset includes a blended data matrix and an unblended data matrix.

11- The authors say the misfit doesn’t change after 40 epochs. It seems they reach a local minimum much earlier.

Response: In this paper, the experimental content is added, and the analysis method of training results is also optimized.Please see “Data vadidation-Test results”section of the revised paper.

12- What is figure 8, 9. No description. No interpretation.

Response: In this paper, the experimental content is added, and the analysis method of training results is also optimized.Please see “Data vadidation-Test results”section of the revised paper.

13- In conclusions the authors say: “Consequently, traditional filtering algorithms employed in seismic data processing fail to yield satisfactory outcomes, particularly in scenarios where ambient noise is prevalent.” Where is this information in the paper? I don’t see any comparison. Benchmark method or traditional method.

Response: We have modified and expanded the conclusion sectionto provide a more comprehensive interpretation of our results and their implications for the field.

is modified to be more appropriate for the design and implementation of the aliasing separation model based on the improved googlenet network

14- Again, in conclusion, the authors compare the performance of their application with other algorithms that were not shown.

Response :In this revised paper, the comparative deblending results with AlexNet, VGG-16,VGG-19,U-Net，GoogLeNet are added.

15- In addition to these comments that are summarized the language of the paper is significantly poor. Some of the phrases are misleading and some can not be understood.

Response: we recheck syntax and description, hope to make the paper more clear and easy to understand.

---

## [Decision Letter · Decision Letter 1]

17 Apr 2024

PONE-D-24-02777R1A blended wavefield separation method for seismic exploration based on improved GoogLeNetPLOS ONE

Dear Dr. Gan,

Thank you for submitting your manuscript to PLOS ONE. After careful consideration, we feel that it has merit but does not fully meet PLOS ONE’s publication criteria as it currently stands. Therefore, we invite you to submit a revised version of the manuscript that addresses the points raised during the review process.

The reviewers raised major comments that need to be addressed in the revised version of the manuscript.

We look forward to receiving your revised manuscript.

Kind regards,

Alberto Marchisio

Academic Editor

PLOS ONE

Reviewers' comments:

Reviewer's Responses to Questions

**Comments to the Author**

1. If the authors have adequately addressed your comments raised in a previous round of review and you feel that this manuscript is now acceptable for publication, you may indicate that here to bypass the “Comments to the Author” section, enter your conflict of interest statement in the “Confidential to Editor” section, and submit your "Accept" recommendation.

Reviewer #2: (No Response)

Reviewer #3: (No Response)

Reviewer #4: (No Response)

Reviewer #5: All comments have been addressed

2. Is the manuscript technically sound, and do the data support the conclusions?

Reviewer #2: Partly

Reviewer #3: Partly

Reviewer #4: No

Reviewer #5: Yes

3. Has the statistical analysis been performed appropriately and rigorously? 

Reviewer #2: N/A

Reviewer #3: No

Reviewer #4: No

Reviewer #5: Yes

4. Have the authors made all data underlying the findings in their manuscript fully available?

Reviewer #2: Yes

Reviewer #3: Yes

Reviewer #4: Yes

Reviewer #5: Yes

5. Is the manuscript presented in an intelligible fashion and written in standard English?

Reviewer #2: Yes

Reviewer #3: Yes

Reviewer #4: No

Reviewer #5: Yes

6. Review Comments to the Author

Reviewer #2: (No Response)

Reviewer #3: Dear Authors,

Thank you for the revised manuscript. It has been partially improved, but many of the previous concerns were not properly addressed. Still, I think the manuscript requires significant revisions. I summarize some comments as follows:

1- There is still no proper data section. The readers cannot be expected to go to the SEG wiki, find the data manual from PGS, and understand how the data were acquired and what the details are.

2- In Figures 5 and 6, what is the value (V)? Do you mean amplitude?

3- In your response, you mentioned that the value in equation 8 corresponds to one. You did not provide further clarification. If it is one, then why is it there?

4- The authors keep mentioning theoretically deblended data. What do you mean by theoretical blended data? Do you mean the benchmark deblended data by PGS? You have to mention how it was obtained.

5- Stressing on the data, one after reading the manuscript will not understand if the data are synthetic or real data. This important information should be embedded in the manuscript.

Reviewer #4: The novelty is incremental. Most of the spce is given to the well known facts and thery. ai is mentioned that imporved goolgenet is used but the improvement is not obvious. Its just using neural network to carry the already well studied deblending task.

Reviewer #5: Dear Editor, Dear Authors,

I did not review the manuscript in the first round; instead, my assessment is based on analyzing the responses to other reviewers' comments from the initial round. I noticed that the authors have made substantial changes compared to the original version, and their responses in the rebuttal letter are generally sound. However, I noticed a minor issue where a few responses appeared to be verbatim repetitions and, in some cases, not directly addressing the specific concerns raised by the reviewers. Aside from this observation, I have no further comments and recommend the manuscript for publication.

Best regards,

Michal Chamarczuk

7. PLOS authors have the option to publish the peer review history of their article (what does this mean?). If published, this will include your full peer review and any attached files.

Reviewer #2: No

Reviewer #3: No

Reviewer #4: No

Reviewer #5: No

---

## [Author Response · Author response to Decision Letter 1]

19 Apr 2024

Dear Alberto Marchisio and Reviewers:

Thank you very much for your careful review of my manuscript and for providing valuable feedback. Your comments have been extremely helpful in improving the quality of the paper. Below, I have addressed each of the reviewers’ comments in detail:

Reviewer #3 Comments

1- There is still no proper data section. The readers cannot be expected to go to the SEG wiki, find the data manual from PGS, and understand how the data were acquired and what the details are.

Response: Upon careful consideration, We added a [Data availability and access] section to the article. In this section, we describe the source of the data and related details.

2- In Figures 5 and 6, what is the value (V)? Do you mean amplitude?

Response:This “value (V)” represents the true value of the data, and the ' V ' in parentheses is the unit of the true value, volt.

3- In your response, you mentioned that the value in equation 8 corresponds to one. You did not provide further clarification. If it is one, then why is it there?

Response: In this paper, seismic data is processed as a special grayscale image, and seismic data is a data with a large dynamic range, and the “value “ in equation 8 needs to be determined according to the maximum and minimum values of the data to be processed. The range of it is also mentioned in equation 8 , which is 1-255, and the value 1 is selected this time. In other application scenarios, other values in the range [1,255] may be selected according to the data characteristics.

4- The authors keep mentioning theoretically deblended data. What do you mean by theoretical blended data? Do you mean the benchmark deblended data by PGS? You have to mention how it was obtained.

Response:metion in section[Data availability and access]. The theoretically deblended data used in this paper were obtained using a deblending method that removes simultaneous source interference by utilizing shot delays in the time-delay matrix to locate and remove unwanted shots from the data matrix, which is also provided in the PGS package.

5- Stressing on the data, one after reading the manuscript will not understand if the data are synthetic or real data. This important information should be embedded in the manuscript.

Response: The blended dataset supporting the findings of this study was the real data of seismic survey provided by PGS and is available for public use.

Reviewer #4 Comments

The novelty is incremental. Most of the spce is given to the well known facts and thery. ai is mentioned that imporved goolgenet is used but the improvement is not obvious. Its just using neural network to carry the already well studied deblending task.

Response: This article innovatively constructs a mathematical model for the separation of blended wavefields based on deep learning theory, addressesing the ill-posed equation solving problem in the separation of blended wavefields. Furthermore, according to the characteristics of seismic data, the effectiveness of the proposed model is validated based on an improved GoogLeNet network. Compared to several other commonly used deep learning networks, the enhanced GoogLeNet not only achieves better deblending effects but also significantly reduces the hardware requirements. The authors consider this approach a significant advancement in the application of AI methods in the field of seismic exploration.

Reviewer #5 Comments

I did not review the manuscript in the first round; instead, my assessment is based on analyzing the responses to other reviewers' comments from the initial round. I noticed that the authors have made substantial changes compared to the original version, and their responses in the rebuttal letter are generally sound. However, I noticed a minor issue where a few responses appeared to be verbatim repetitions and, in some cases, not directly addressing the specific concerns raised by the reviewers. Aside from this observation, I have no further comments and recommend the manuscript for publication.

Response:Thank you for your thorough evaluation of the manuscript, and we have further addressed the reviewers’ concerns.

Thanks for your time and consideration of our manuscript. We look forward to hearing from you regarding the outcome of the review process.

Sincerely,

ZhiQiang Gan，Xiang-E Sun

Yangtze University

---

## [Decision Letter · Decision Letter 2]

8 May 2024

A blended wavefield separation method for seismic exploration based on improved GoogLeNet

PONE-D-24-02777R2

Dear Dr. Gan,

We’re pleased to inform you that your manuscript has been judged scientifically suitable for publication and will be formally accepted for publication once it meets all outstanding technical requirements.

Kind regards,

Alberto Marchisio

Academic Editor

PLOS ONE

Additional Editor Comments (optional):

Reviewers' comments:

Reviewer's Responses to Questions

**Comments to the Author**

1. If the authors have adequately addressed your comments raised in a previous round of review and you feel that this manuscript is now acceptable for publication, you may indicate that here to bypass the “Comments to the Author” section, enter your conflict of interest statement in the “Confidential to Editor” section, and submit your "Accept" recommendation.

Reviewer #4: All comments have been addressed

Reviewer #5: All comments have been addressed

2. Is the manuscript technically sound, and do the data support the conclusions?

Reviewer #4: Yes

Reviewer #5: Yes

3. Has the statistical analysis been performed appropriately and rigorously? 

Reviewer #4: Yes

Reviewer #5: Yes

4. Have the authors made all data underlying the findings in their manuscript fully available?

Reviewer #4: No

Reviewer #5: Yes

5. Is the manuscript presented in an intelligible fashion and written in standard English?

Reviewer #4: Yes

Reviewer #5: Yes

6. Review Comments to the Author

Reviewer #4: (No Response)

Reviewer #5: I think the the authors addressed the further comments substantially. In my opinion this manuscript should be accepted with its current version. Congratulations to all the authors!

7. PLOS authors have the option to publish the peer review history of their article (what does this mean?). If published, this will include your full peer review and any attached files.

Reviewer #4: No

Reviewer #5: No

---

## [Editor Report · Acceptance letter]

26 May 2024

PONE-D-24-02777R2 

PLOS ONE

Dear Dr. Gan, 

I'm pleased to inform you that your manuscript has been deemed suitable for publication in PLOS ONE. Congratulations! Your manuscript is now being handed over to our production team.

Kind regards, 

on behalf of

Dr. Alberto Marchisio 

Academic Editor

PLOS ONE